# Manufacturing of Complex Silicon–Carbon Structures: Exploring Si_x_C_y_ Materials

**DOI:** 10.3390/ma15103475

**Published:** 2022-05-12

**Authors:** Skyler Oglesby, Sergei A. Ivanov, Alejandra Londonõ-Calderon, Douglas Pete, Michael Thompson Pettes, Andrew Crandall Jones, Sakineh Chabi

**Affiliations:** 1Department of Mechanical Engineering, University of New Mexico, Albuquerque, NM 87131, USA; oglesbys@unm.edu; 2Materials Physics and Application Division, Los Alamos National Laboratory, Los Alamos, NM 87545, USA; ivanov@lanl.gov (S.A.I.); alondono@lanl.gov (A.L.-C.); pettesmt@lanl.gov (M.T.P.); acj@lanl.gov (A.C.J.); 3Sandia National Laboratories, Albuquerque, NM 87123, USA; dvpete@sandia.gov

**Keywords:** silicon carbide, Si_x_C_y_, lightweight materials, nanosheets, foams

## Abstract

This paper reports on the manufacturing of complex three-dimensional Si/C structures via a chemical vapor deposition method. The structure and properties of the grown materials were characterized using various techniques including scanning electron microscopy, aberration-corrected transmission electron microscopy, confocal Raman spectroscopy, and X-ray photoelectron spectroscopy. The spectroscopy results revealed that the grown materials were composed of micro/nanostructures with various compositions and dimensions. These included two-dimensional silicon carbide (SiC), cubic silicon, and various SiC polytypes. The coexistence of these phases at the nano-level and their interfaces can benefit several Si/C-based applications ranging from ceramics and structural applications to power electronics, aerospace, and high-temperature applications. With an average density of 7 mg/cm^3^, the grown materials can be considered ultralightweight, as they are three orders of magnitude lighter than bulk Si/C materials. This study aims to impact how ceramic materials are manufactured, which may lead to the design of new carbide materials or Si/C-based lightweight structures with additional functionalities and desired properties.

## 1. Introduction

Silicon and carbon are among the most critical research areas in nanotechnology, and any advancement in these areas can impact several scientific fields. For example, the discovery of low-dimensional carbon materials, such as graphene and carbon nanotubes, have had a massive impact on modern technology development such as wearable electronics, transparent flexible devices, ultrasensitive sensors, and atomically thin separation membranes [1,2,3,4]. On the other hand, silicon is the most widely used material for semiconductor devices. Silicon-based semiconductor technology has produced numerous breakthroughs in electronics. However, with Moore’s law approaching its end, the search for a semiconducting successor to Si has already begun [5,6,7,8].

The combination of Si and C (i.e., SiC) is yet another important and essential structure with various uses including high power electronics and quantum information processing [9,10]. As a wide bandgap semiconductor, SiC is a leading material for high-voltage/frequency power electronics. SiC is also useful for many other vital industries and technologies such as aerospace, e.g., space mirrors, photonics, quantum information processing, and airframes (Figure 1). This broad market for SiC directly results from its unique properties such as a large bandgap, high thermal stability and conductivity, chemical stability, and mechanical properties [11,12].

Furthermore, silicon, carbon, and SiC can show different properties and functionalities useful for various applications depending on their dimensions [13,14,15,16,17,18,19]. For example, in our previous work [20], we reported the synthesis of graphene foam and three-dimensional graphene composites that showed remarkable electrochemical and mechanical properties when used for supercapacitor applications. After ten thousand charge–discharge cycles, the graphene foam composite supercapacitors showed 100% capacity retention. Such a high level of performance has never been reported or observed with other forms of carbon materials such as activated carbon or carbon black.

We also made 1:1 SiC foam [21] with interesting mechanical properties such as responding to mechanical load via elastic–plastic deformation and a very high strength-to-weight ratio. These SiC foams recovered their original shape, to a great degree, after being exposed to excessive compression loads. This ductile-like deformation and recoverability has also been reported in ceramic nanolattices [22]. Unlike conventional ceramics structures that have limited stretchability and are unsuitable for many engineering applications, ductile ceramics can enable many applications.

These interesting findings strongly confirm that one efficient approach to designing and engineering materials with new properties and enhanced functionalities is by manipulating the dimensions of the material. The same materials can show different properties depending on their dimensions and size. For example, a brittle–ductile-like transition has been reported for various 3D nanolattices due to the manipulation of their sizes [23].

Despite the proven effectiveness, dimensionality may not be the only/best way to manipulate the properties of Si/C materials. Another potential approach, which is the primary motivation behind this research, is to play with the composition and atomic percentage of Si/C materials and design Si_x_C_y_ materials. Although a 1:1 composition or SiC is the most stable form of Si_x_C_y_, other compositions, for example, SiC_3_ or Si_3_C, are also found to be energetically favorable [13,16,24].

Si_x_C_y_ materials have never been thoroughly explored before in terms of experimental works. Many research groups have heavily studied Si-C composites, Si-SiC composites, and C-SiC composites. Nevertheless, these few examples of the possible Si_x_C_y_ structures are part of a massive family of complex Si/C systems. Depending on the composition (whether Si or C is dominant), these materials may behave as electrical conductors or semiconductors and as brittle or ductile material or show a new set of properties and functionalities [25,26,27,28,29]. They may also have different micro/nanostructures, interfaces, defect levels and, thus, different properties and applications will exist. 

With further exploration of the composition and more focus on desired synthesis parameters, many attributes and applications could be uncovered and enabled by Si_x_C_y_ material. Herein, we aimed to explore the feasibility of synthesizing Si_x_C_y_ materials. 

## 2. Results and Discussion

The complex 3D Si/C structures, or simply Si_x_C_y_ foams, were formed via a chemical vapor deposition (CVD) reaction between an ultralightweight graphene foam (GF) and an excessive amount of Si precursor (SiO). Appendix A shows images of the graphene precursor and the produced foam. A detailed description of the synthesis procedure can be found in the experimental section and Appendix A. With a calculated density in the range of 5–9 mg/cm^3^, the grown foam was three orders of magnitude lighter than bulk Si and SiC material at room temperature and atmospheric pressure. 

The physical and chemical structures of the grown materials were characterized using several techniques. Figure 2 displays results from scanning electron microscopy (SEM), energy-dispersive spectroscopy (EDS), and X-ray diffraction (XRD). The SEM results (Figure 2a) show that, like their graphene precursor, the grown foams had a porous 3D structure with a pore size of nearly 250 μm. Figure 2b is a high-magnification SEM image showing the morphology and texture of the grown foams. Micro/nanowires can also be detected in another SEM image of the foam (Figure 2c). The formation and densities of these nanowires could be attributed to traces of nickel foam, which was used as the beginning substrate of the GF [30]. However, all our characterization tests confirmed successful etching of nickel foam in hydrochloric acid, as the metal was not detected in the structural tests. The elemental composition of the synthesized materials was investigated using EDS.

The EDS spectrum of a freshly prepared foam (Figure 2d) revealed the presence of oxygen, silicon, and carbon. The presence of an oxygen peak indicates the formation of undesirable SiO_x_, which happens during the synthesis process. To address this issue, a hydrofluoric acid (HF) wet etch followed by a hydrochloric acid (HCl) ionic clean was used to eliminate this oxide layer. The EDS results for the HF-etched foams (Figure 2e) confirmed the elimination of the oxide layer from the post-etching measurements. EDS results from various samples also revealed that the foam contains Si and C in a molar ratio of 4:1. These results suggest that the grown Si_x_C_y_ foams were Si-rich structures. All further characterizations reported below were conducted for the HF-etched samples unless otherwise stated.

The structure and morphology of the grown material was characterized by utilizing powder XRD (Figure 2f), which revealed the presence of at least three different crystalline phases. This was expected because of the high Si:C ratio. The various XRD peaks can be indexed to Si and SiC crystal structures. Peaks that surfaced at 28, 47, and 56° were indexed to Si. The presence of α-SiC and β-SiC polymorphs were also evident. Both phases can be indexed to the utmost intense peak (35.5°) and at 60 and 72°. The small peak at 41° corresponded to cubic 3C-SiC, and the peak at 33° was typical with hexagonal 2H-SiC. The absence of a carbon peak suggests the complete transformation of GF into Si_x_C_y_ foam throughout the CVD and etch procedure. As it became clear that the produced foam was Si-rich Si_x_C_y_ (as per EDS and XRD), it was necessary to study the foam’s nanostructure through TEM.

Figure 3 shows TEM images of the foams together with elemental X-ray mapping, selected area electron diffraction (SAED), and high-angle annular dark-field scanning transmission electron microscopy (HAADF-STEM). Two types of nanostructures were identified: nanosheets and nanowires. Figure 3a shows a TEM image of a nanosheet with a nanowire coexisting and overlapping a portion of it (bottom right corner). This provides good contrast between these nanostructures. The color contrast in several regions of the nanosheets was related to the thickness change and folded edges. The thinner regions of the nanosheets appeared more transparent.

Figure 3b,c show TEM images from individual nanowires and nanosheets. Although carbon-based TEM grids were used for imaging, the elemental mapping results still deserve close attention. The elemental mapping confirms that the grown materials were Si-rich, as the density of Si particles (green) was much higher than that of carbon (red) (Figure 3d–f). High-resolution TEM (HRTEM) images of the nanosheets confirm the highly crystalline nature of the nanostructures (Figure 3g,h). These nanosheets were primarily composed of hexagonal SiC.

Figure 3h offers a closer look at the atomic lattice and orientation of the formed nanosheets and the associated SAED (Figure 3i), revealing the coexistence of different lattices. Interestingly, a hexagonal lattice can also be detected and attributed to the SiC nanosheets. Mono/few-layer SiC, or simply 2D SiC, is emerging as a new material with distinctive structural, optical, and electrical properties. More information on these nanosheets and two-dimensional SiC was recently published elsewhere [31]. 

The HRTEM image shown in Figure 3j provides a closer look at the SiC nanowires. The overlapping structures are visually different. The wire with a larger diameter possessed a mostly uniform structure of β-SiC 3C established by the interplanar distance measurement of 0.25 nm along the direction of growth. The thinner wire in Figure 3j is visibly twinned, as its atomic arrangement lies within the twinned layers shown in Figure 3k. The atomic arrays had ABC stacking and a cubic structure. There were stacking faults perpendicular to the wire’s axis, denoted by “sf” in the figure. The HRTEM results, along with the SAED result (Figure 3l), confirm that these nanowires were made up of the β-SiC 3C polytype. More TEM images of the grown materials are presented in Appendix A. Although the HRTEM images focused on the SiC phase, both elemental mapping and the SAED results indicate that the grown materials were multiphase.

X-ray photoelectron spectroscopy (XPS) was used to investigate the bonding structure and, once again, verified the composition of the grown materials. Figure 4a,b show the XPS spectrum of the produced materials. In the C 1s graph, the intense peak at 283 eV can be assigned to the SiC structure. The peak at 285 eV was linked to sp^2^/sp^3^ carbon–carbon bonding. In the Si 2p graph, Figure 4b, the peak at 99.5 eV can be assigned to Si 2p bonding in Si-Si structures. The peak at 100.3 eV corresponded to SiC, while the very small peak at 101 eV can be indexed to SiO. Thus, in addition to SiC and Si, small amounts of SiO could also be detected in the grown materials.

Figure 4c shows the Raman behavior of the grown foams under different excitation wavelengths. All spectra had a peak at approximately 780 cm^−1^, corresponding to the transverse optical (TO) mode of SiC, which indicates that SiC was the dominant phase in the tested samples. The Si peak could be detected at 520 cm^−1^ using green and red laser light (488.0 and 532.3 nm, respectively). In both cases, the Si peak appeared at 510 ${\mathrm{cm}}^{-1}$, different from the expected 520 ${\mathrm{cm}}^{-1}$ peak. This shift in the peak position can be attributed to potential residual strain within the sample/substrate.

It is worth noting that the SiC phase of the grown materials did not show a typical Raman signature of any known SiC polytype. For example, as shown in Appendix A, cubic and hexagonal SiC polytypes had two distinct Raman peaks; the TO peak at approximately 790 cm^−1^ and the longitudinal optical (LO) peak at approximately 960 cm^−1^. In the case of 2H-SiC, two TO peaks existed (the small shoulder to the left side of the TO band). Most of our grown materials did not show a noticeable LO peak. However, we observed an anticorrelation between the LO peak of the SiC phase and the Raman peak of silicon: the stronger the Raman peak of Si implied a weaker LO peak in the SiC phase (see Figure 4). These results further confirm the complex multiphase nature at the atomic level and that these materials are not simply a composite of SiC and silicon. Instead, they reveal the potential of varying the composition of Si/C materials at the atomic level, leading to the formation of Si_x_C_y_ structures with controlled composition and desired properties.

The Si_x_C_y_ foam’s thermal behavior was inspected using thermogravimetric analysis (TGA). Figure 4d shows TGA data on the sample under nitrogen (blue line) and air (red line) environments. The sample was heated to 700 °C at 30 °C/min with subsequent heating to 1500 °C at 10 °C/min under a flow of nitrogen or air at 120 mL/min. The sample remained thermally and chemically stable up to 1500 °C upon heating under N_2_, whereas heating the sample under air resulted in oxidation at temperatures above 900 °C. The XRD profile of the sample after heating under N_2_ coincided with that of the original sample (Figure 4e). XRD also confirmed the sample’s oxidation upon heating in air (Figure 4f), with the formation of ${\mathrm{SiO}}_{2}$.

These findings indicate that the grown materials are beneficial for high-temperature applications where both thermal and chemical stability are desired. Although this report does not claim the systematic development of Si_x_C_y_ materials with precise composition and controlled structure/properties, it did succeed in making complex ultralightweight Si/C 3D structures in which the elemental composition of silicon exceeded that of carbon. Our findings suggest that the composition of Si_x_C_y_ materials may be manipulated by experimenting with different weight ratios of silicon and carbon precursors during the CVD process. The dominant growth mechanism in the synthesis process is the solid (graphene foam)–vapor (SiO) reaction. It is anticipated that this work will inspire further experimental investigation into Si_x_C_y_ materials. 

## 3. Conclusions

This work demonstrated the successful formation of complex multiphase Si/C materials by a CVD method using a graphene precursor. The grown ultralightweight 3D materials were composed of Si phases, different polytypes of SiC, and two-dimensional SiC nanosheets. The coexistence of these phases and micro/nanostructures was a direct result of the synthesis approach used in this work. The grown materials showed interesting Raman and XPS behavior and enhanced thermal and chemical stability, different from that of pure Si or pure SiC. Our findings suggest that a new and prominent family of Si/C materials with unprecedented properties can be achieved by successfully manipulating the composition and dimension of these materials. With additional investigation and mending of Si_x_C_y_ materials and Si/C compositions, revolutionary developments can be made to modern technologies.

## 4. Experimental Procedure

Fabrication: The Si-rich foam was formed via a CVD reaction between graphene foam (2 cm × 2 cm × 1.6 mm thickness) and SiO (1 g, Sigma–Aldrich, St. Louis, MO, USA). Detailed information about the synthesis of the graphene foam (density of 3 mg/cm^3^) can be found in our previous work [20]. The system was brought from room temperature to a final 3 h dwelling temperature of 1450 °C. The experiment was performed under argon gas, 200 sccm. During the synthesis, the SiO vapor formed at high temperature traveled upstream to reach the graphene foam and reacted with it. The flow rate was reduced to 50 sccm during the cooling process. All foams were etched in a dilute 48% HF for 3 h at room temperature to eliminate any unreacted SiO or potential SiOx layer. The sample was then rinsed with a 1M HCl solution, followed by a drying stage before being characterized.

Measurements: SEM was used for analyzing the 3D structure of the grown materials. Structural visualization and precise measurements were made using the Nova NanoSEM 450, a two-FEI field-emission source SEM operating at 15 kV. EDS was conducted to determine the atomic composition of our material and was performed inside the SEM. A Rigaku Ultima III powder diffractometer was utilized for XRD measurements. The Titan environmental TEM equipped with a Gatan K2-IS camera was used for HRTEM and SAED measurements, while the JOEL NEOARM (200 kV) aberration correction STEM with a full-spectral EDS was used for electron mapping and other HRTEM imaging. For TEM sample preparation, the crushed Si/C foam was ultrasonically dispersed in acetone for 10 min, and then the suspension was dropped onto a holey carbon-coated copper grid (300 mesh, Agar).

Raman measurements were conducted using a Horiba LabRAM HR Evolution high-resolution confocal Raman microscope fitted with volume Bragg gratings. Spectral calibration was performed using the 1332.5 cm^−1^ band of a Type IIa diamond [32]. XPS measurements were performed using a Kratos AXIS ULTRA X-ray Photoelectron Spectrometer. The simultaneous TGA/DSC Netzsch STA 449 F1 Jupiter was used for the TGA analysis. 

## Figures and Tables

**Figure 1 materials-15-03475-f001:**
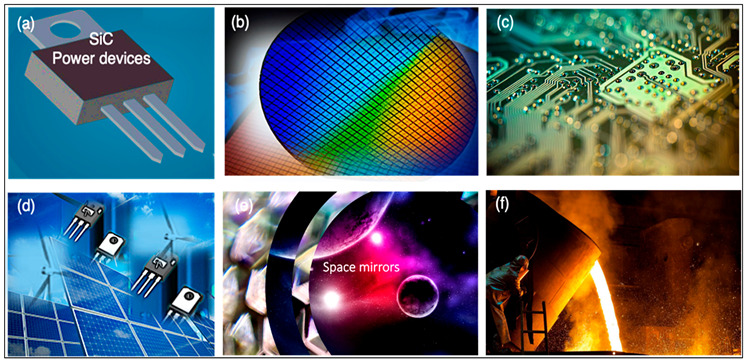
An overview of SiC applications: Examples include: (**a**) power semiconductor devices; (**b**) wafers; (**c**) integrated circuits; (**d**) renewable energies; (**e**) space mirrors; (**f**) high-temperature applications. SiC materials have also been used widely for quantum technologies, nuclear energy, aviation, abrasives, and several structural applications.

**Figure 2 materials-15-03475-f002:**
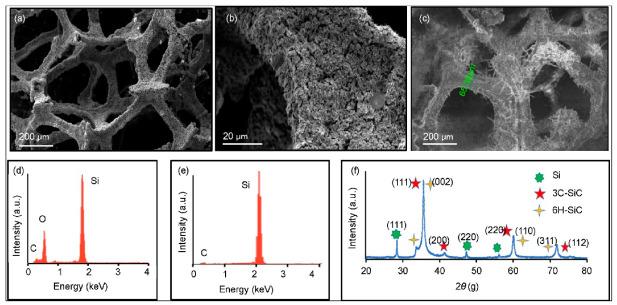
Morphological and elemental characterization of the grown foam: (**a**–**c**) SEM images; (**d**,**e**) EDS data taken before and after an HF etch; (**f**) XRD pattern of the HF-etched foam with hkl indices.

**Figure 3 materials-15-03475-f003:**
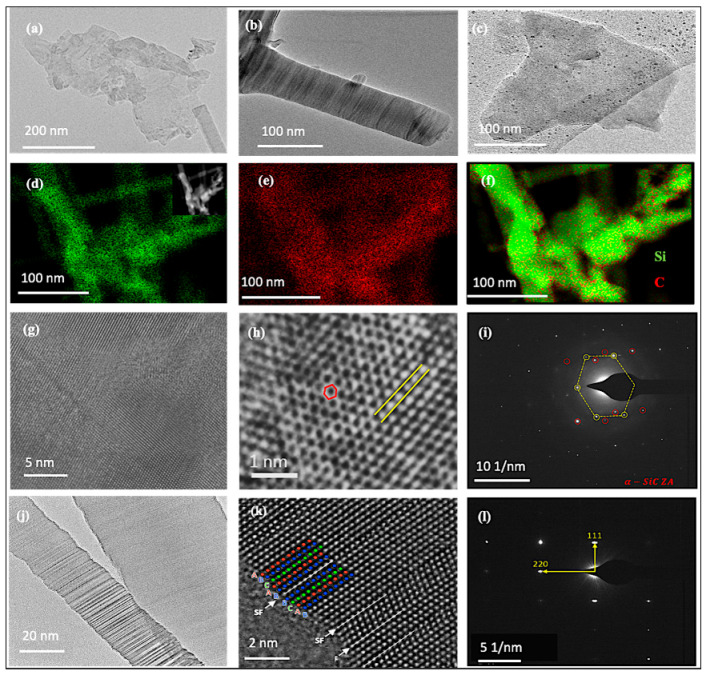
TEM results: (**a**–**c**) TEM images revealing that the grown 3D structures were composed of nanosheets and nanowires; (**d**–**f**) elemental X-ray mappings corresponding to the STEM-HAADF images shown in the inset of image (**d**); (**g**) HRTEM image showing the highly crystalline nature of the grown material; (**h**,**i**) HRTEM images of a different structure and the associated selected area electron diffraction pattern; (**j**,**k**) HRTEM images of a grown twinned nanowire (left) and (**l**) its indexed selected area electron diffraction pattern.

**Figure 4 materials-15-03475-f004:**
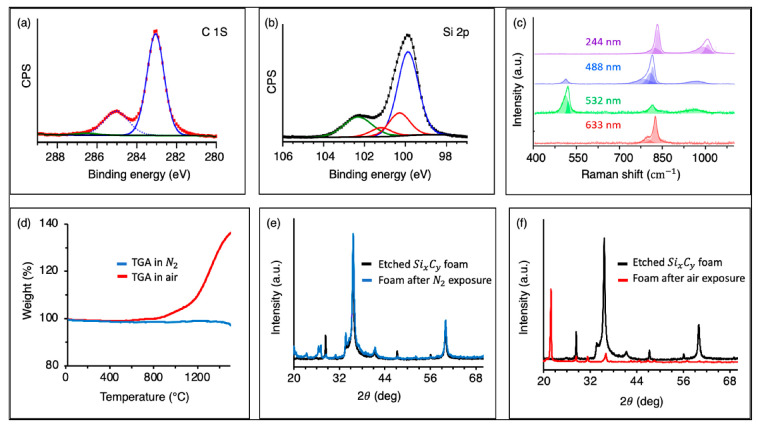
(**a**,**b**) XPS surface survey spectrums from C1s and Si 2p signals, respectively; (**c**) Raman results using different excitation energies; (**d**) TGA of the foam in nitrogen and air atmospheres; (**e**,**f**) XRD results of the foam overlapped with results after TGA experiments.

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
