# Peer review of "Manufacturing of Complex Silicon–Carbon Structures: Exploring SixCy Materials"

_materials, 2022, doi:10.3390/ma15103475_

Round 1

Reviewer 1 Report

This work demonstrated the successful formation of complex multiphase Si/C materials by CVD method using graphene precursor and SiO, and the structure and properties of the grown materials were characterized using various techniques. This article would be published by Materials, although some questions need to be answered:

(1) Was the method of producing SixCy materials by CVD reaction between SiO and C materials (not only graphene) original or the first time in this work? It may be illustrated in the Introduction secton.

(2) The grown foams are Si-rich, is it reacted with HF?

Author Response

Reviewer 1

Comments and Suggestions for Authors

This work demonstrated the successful formation of complex multiphase Si/C materials by CVD method using graphene precursor and SiO, and the structure and properties of the grown materials were characterized using various techniques. This article would be published by Materials, although some questions need to be answered:

RE: Thank you for the recommendation.

  • Was the method of producing SixCymaterials by CVD reaction between SiO and C materials (not only graphene) original or the first time in this work? It may be illustrated in the Introduction secton.

RE: Thank you for the comment. It is not original to this work. We used this method in the past (2015), to make SiC foam. Here is the link to our previous publication. https://doi.org/10.1021/acsnano.5b05533

  • The grown foams are Si-rich, is it reacted with HF?

RE: As mentioned in the manuscript, we used HF etching to get rid of unreacted SiO, or any SiOx. We added this info to the experimental section.  However, the grown Si-rich foam, itself is very stable towards HF-etching.

Reviewer 2 Report

The article entitled "Manufacturing of Complex Silicon-Carbon Structures: Exploring SixCy Materials" demonstrates a novel approach for ultralight SiC foam synthesis by CVD method. The authors of the manuscript present a continuation of the work, the results of which were previously published [Ref. 32]. In my opinion, both publications, reviewed and earlier one, form a logical whole from the synthesis of 2D material to the 3D version. The presented results are remarkable and have the features of innovative work. Therefore, I recommend the work for publication, but because of a lot of minor flaws and inaccuracies, I suggest major revision of the manuscript.

My comments are as follows:

  1. In the introductory part, the authors revise previous achievements in a similar field and cite many works, but a lot of them were published over 10 years ago. Among them, there is even a review from the last century [Ref. 10]. I would suggest a more up-to-date review.
  2. In my opinion, Figure 1 is completely unnecessary, as it does not add any substantive content to the article.
  3. It seems more sensible to use Figure S2 as an addition to Chapter 2, instead of including it in the supplementary part.
  4. In lines 103 and 104, the authors claim that there are no traces of Ni, but this is not due to the EDS results shown as the Ni K-alpha has a much higher energy than the maximum value on the axis (7.47 KeV).
  5. In line 126, the authors argue that the absence of graphene reflections confirms complete conversion to carbide. I note that graphene as a 2D material does not have reflections in the powder XRD, but in the form of an oxide, the reflections should appear at 10 degrees, and no measurements were made in this angular range.
  6. In line 149/150 the authors interpret the HRTEM images claiming that there is a hexagonal crystal form of SiC. Is it possible to develop a cubic SiC nanosheet, but with an exposed (111) surface?
  7. The diffractograms (Figure 4 e, f) would be clearer if the "etched" and "N2 exposed" pattern were shifted and not overlapped.
  8. I have the impression that the authors approach the manuscript in a sloppy manner, as there are many editorial errors. Starting from the line with the authors names and ending with the lack of spaces in the descriptions of the figures. I propose a very thorough proofreading of the entire text.

Author Response

Reviewer 2

Comments and Suggestions for Authors

The article entitled "Manufacturing of Complex Silicon-Carbon Structures: Exploring SixCy Materials" demonstrates a novel approach for ultralight SiC foam synthesis by CVD method. The authors of the manuscript present a continuation of the work, the results of which were previously published [Ref. 32]. In my opinion, both publications, reviewed and earlier one, form a logical whole from the synthesis of 2D material to the 3D version. The presented results are remarkable and have the features of innovative work. Therefore, I recommend the work for publication, but because of a lot of minor flaws and inaccuracies, I suggest major revision of the manuscript.

 RE: Thank you for the recommendation.

My comments are as follows:

  1. In the introductory part, the authors revise previous achievements in a similar field and cite many works, but a lot of them were published over 10 years ago. Among them, there is even a review from the last century [Ref. 10]. I would suggest a more up-to-date review.

RE: Thanks for the comment. We add more recent references in the introduction part. We replaced ref 10 with a new one. Related references are highlighted in the introduction part ( 1st and 2nd paragraph).

  1. In my opinion, Figure 1 is completely unnecessary, as it does not add any substantive content to the article.

Re: This figure illustrates the applications of SiC, and how broad, and important the applications are. We thought, it is very relevant.

  1. It seems more sensible to use Figure S2 as an addition to Chapter 2, instead of including it in the supplementary part.

Re: Thanks for the comment. But, given the content of the figure ( e.g. clouds ) we rather to present it in the supporting information

  1. In lines 103 and 104, the authors claim that there are no traces of Ni, but this is not due to the EDS results shown as the Ni K-alpha has a much higher energy than the maximum value on the axis (7.47 KeV).

RE: Thanks for the comments. Our XRD, and Raman, and TEM results also confirm the absence of Ni.

In line 126, the authors argue that the absence of graphene reflections confirms complete conversion to carbide. I note that graphene as a 2D material does not have reflections in the powder XRD, but in the form of an oxide, the reflections should appear at 10 degrees, and no measurements were made in this angular range.

RE: We replaced “confirms” with “suggests”. We also replaced graphene peak with carbon peak.  We used 1.6 mm thick graphene foam, instead of monolayer graphene as a precursor. Like graphite, graphene foam shows peak in XRD.

  1. In line 149/150 the authors interpret the HRTEM images claiming that there is a hexagonal crystal form of SiC. Is it possible to develop a cubic SiC nanosheet, but with an exposed (111) surface?

RE: According to theoretical studies, monolayer SiC adopts hexagonal structure. As it is the most stable form.

  1. The diffractograms (Figure 4 e, f) would be clearer if the "etched" and "N2 exposed" pattern were shifted and not overlapped.

RE: Thanks for the comment.   We made minor revision to this figure, by changing the axis scale (minor unites). The purpose here is to compare the two foams, as such we overlapped the graphs.

  1. I have the impression that the authors approach the manuscript in a sloppy manner, as there are many editorial errors. Starting from the line with the authors names and ending with the lack of spaces in the descriptions of the figures. I propose a very thorough proofreading of the entire text.

RE: Done.  New changes are highlighted throughout the manuscript. Thank you for the comment.

Reviewer 3 Report

The work is scientifically sound, and while there is new information here, the methodology novelty is somewhat adequate.

In a present state the manuscript submitted is adequate and to be recommended for publication with minor correction. 

Listed of the corrections and question:

  1. Reference no. 23 should be changed or must be refer to article/journal and not http/website.

Author Response

Reviewer 3

Comments and Suggestions for Authors

The work is scientifically sound, and while there is new information here, the methodology novelty is somewhat adequate.

In a present state the manuscript submitted is adequate and to be recommended for publication with minor correction. 

Re: Thanks for the recommendations

Listed of the corrections and question:

  1. Reference no. 23 should be changed or must be refer to article/journal and not http/website.

RE: Done. We removed that figure, and thus the associated reference, and prepared a new figure using a software ( figure 1e).

Round 2

Reviewer 2 Report

Thank you for applying some of my suggestions. I have no more comments.